# Relationship between Competency to Consent to Treatment and Psychological Well-Being: Mediating Effect of Empowerment and Emotion

**DOI:** 10.3390/ijerph18158170

**Published:** 2021-08-02

**Authors:** Yeun-Joo Hur, Joon-Ho Park, MinKyu Rhee

**Affiliations:** 1Institute for Human Rights & Social Development, Gyeongsang National University, Jinju 52828, Korea; mommy0311@naver.com; 2Department of Psychology, Gyeongsang National University, Jinju 52828, Korea; jjoon@gnu.ac.kr

**Keywords:** competency to consent to treatment, psychological well-being, empowerment, positive emotion, negative emotion, psychiatry outpatient

## Abstract

This study was conducted to evaluate the competency to consent to the treatment of psychiatric outpatients and to confirm the role of empowerment and emotional variables in the relationship between competency to consent to treatment and psychological well-being. The study participants consisted of 191 psychiatric outpatients who voluntarily consented to the study among psychiatric outpatients. As a result of competency to consent to treatment evaluation, the score of the psychiatric outpatient’s consent to treatment was higher than the cut-off point for both the overall and sub-factors, confirming that they were overall good. In addition, the effect of the ability of application on psychological well-being among competency to consent to treatment was verified using PROCESS Macro, and the double mediation effect using empowerment and emotional variables was verified to provide an expanded understanding of this. As a result of the analysis, empowerment completely mediated the relation between the ability of application and psychological well-being, and the relation between the ability of application and psychological well-being was sequentially mediated by empowerment and emotion-related variables. Based on these findings, the implications and limitations of this study were discussed.

## 1. Introduction

In the 21st century regarded as the ‘era of diversity’, there is an increasing interest in social minorities in the whole society. However, human rights cannot be guaranteed in the absence of a social safety net. Therefore, many countries are trying to improve the sensitivity and judgment of individuals regarding human rights within society through laws and national policies [1]. Although social minority groups are diverse, disabled groups feel the most distant to the general public [2]. People with disabilities occupy the largest proportion among social minority groups [3]. A person with a disability is defined as a person who is severely restricted in daily life or social life over a long period due to a physical or mental disorder [4]. Disabled ones can be classified into physically disabled and mentally disabled according to the cause of the disability, and the human rights problems faced by the two groups differ in scope and degree of discrimination. Specifically, people with physical disabilities frequently experience human rights issues related to physical living conditions such as education, mobility, residence, occupation, leisure, culture, and information access. On the other hand, people with mental disorders and mental disorders are known to be more likely to experience more serious human rights problems such as forced hospitalization and treatment, abuse or neglect, violence, and social stigma. Thus, they have the lowest levels of human rights [5].

For example, when treating a physical condition of disabilities, refusing treatment by subjects does not mean that they lack the competency to consent to treatment. On the other hand, people with mental disorders are judged to have a lack of insight and impairment of cognitive and thinking functions, thus presupposing their ‘inability’ (i.e., they cannot make decisions about treatment or life without a valid evaluation procedure) [6]. In addition, prejudice and unfounded discrimination against mentally ill patients such as dangers and irrecoverable aspects of Korean society deprive psychiatric patients of the “right to decide on their own treatment” differently from those with physical disabilities [7]. This means that even in the field of health, there is a difference in view that considers physical and mental disorders differently [2].

However, the social outlook on people with mental illness has not changed significantly. Since the development of antipsychotic drugs in the 1950s, the concept of outpatient treatment has become common among people with mental disorders. It is known that early detection of mental disorders can sufficiently compensate for the decline in daily function [8]. According to the Korean government’s mental health survey conducted every five years, the rate of participation in mental health service in Korea is about half of that in advanced foreign countries [9]. In addition, it is suggested that the estimated population for the mentally disabled is estimated to be 108,000 [9]. When referring to other data, the lifetime prevalence of mental illness in Korea is high at 25.4% [10]. The reality is that our society still has a prejudice against the mentally ill, making the mentally ill people self-confident in social isolation or reluctant to find a disease because they fear that social disadvantages will occur [11].

The basic problem that presupposes the inability of a person with mental illness arises from the competency to consent to treatment. The judgment that the mentally impaired person’s competence to consent is insufficient leads to a decision on behalf of treatment without consent of the person concerned [6]. The recent trend of mental health in Korea is pursuing the ideology of deinstitutionalization in which living spaces belong to local communities [12]. Unlike when most people with mental disorders lived in hospitals in the past, most people with mental disorders live in local communities. Therefore, it is necessary to accurately identify their competency to consent to ensure that it is appropriate to presuppose incompetence and to have the opportunity to correct stereotypes about them.

In addition, since most people with mental disorders have difficulty in improving their quality of life due to other psycho-social factors, even if symptoms are reduced [13], there is a growing interest in factors that can influence their quality of life. However, the generalized prejudice that mentally ill people are incapacitated (such as no competency to consent) makes it difficult to pay sufficient attention to their quality of life and related variables. Studies examining their competency to consent to treatment and their psychological well-being with related variables are insufficient. There are many studies on the rights of mental patients to understand the predisposition that affects their social rights such as social stigma, discrimination, and decision-making rights [14,15]. However, what humans ultimately want is a happy life [16]. The right to pursue happiness is also a basic right that humans must enjoy. A happy life is a key topic in the recent era. Since goals of various national policies are shifting toward enhancing the happiness of people [17], it is important to examine the relationship between competency to consent and psychological well-being. Attention to these topics is of greater importance for those involved in mental health services in the community. This is because a low level of competency to consent to treatment does not mean no competency to consent just because one sub-capacity is insufficient [6,18]. For those participating in mental health services, it is important to understand which capabilities affect their happy lives to improve their well-being through community programs.

Finally, it is appropriate to understand the influence of empowerment and emotion on the relationship between competency to consent and psychological well-being. Empowerment is highly related to the concept of social participation or adaptation in the aspect of emphasizing control and choice over the environment [19,20]. It is known that mentally ill people often experience emotional problems [21].

In particular, the influence of empowerment is considered very important to the mentally ill. In the concept of empowerment, the area of choice is emphasized, and in positive psychology, the choice is sometimes presented as an important factor in living a prosperous life [20,22]. As a result, empowerment plays an important role in forming a positive sense of self in that the mentally ill, a group of social minorities, can coordinate life and their environment rather than seeing themselves as beneficiaries of social protection [23]. As such, empowerment could affect the psychological well-being of people with mental illness. The influence of emotion-related variables cannot be overlooked, as an individual’s internal factor can affect the psychological well-being of the mentally ill. When considering emotions, the absence of a negative effect does not mean positive effects [24]. Thus, both negative and positive emotions should be focused on. Emotional stability is related to a person’s feelings of happiness, optimism, and mildness [25]. In particular, mentally ill people are more likely to be exposed to stigma or discrimination caused by prejudice and stereotypes. Depression also impairs their competency to consent [26]. Therefore, a vicious cycle of increasing chance of experiencing emotional problems due to social maladjustment of a mentally ill person can be repeated.

### 1.1. Competency to Consent to Treatment

Conceptually, competency to consent to treatment is defined as the ability to understand facts and treat information rationally to evaluate the most conjoined realistically compared to other alternatives [6,18]. It is presented as a combination of cognitive factors, including the following sub-factors: the ability to express, the ability to recognize information about symptoms and treatment methods, the ability of application to apply the provided information to oneself, and the ability of reasoning to logically determine treatment rejection and acceptance. Since the ability to express is a measure of the aspect of expressing the competence to consent to treatment, if the ability to express decreases to a score below the cut-off level, it is considered that there is no competency to consent regardless of scores for the remaining three sub-factors. However, previous studies have argued that if the ability to express the competency to consent to treatment is maintained, the four sub-factors are multi-dimensional. Therefore, they require independent evaluation. It has been asserted that the perfection of one ability should not be regarded as sufficient for other abilities. This means that the imperfection of one ability should not be regarded as a decrease in the overall ability to express.

The right to self-determination in treatment is not limited to people with disabilities, but this is a particularly important aspect for the disabled. Its importance has been emphasized by the “convention on the rights of People with Mental Disabilities.” [27]. For people with disabilities, the ability to express begins with an “informed consent,” which is the basis for treatment ethics. Then, a clear concept of “competency to consent” is defined [28]. In Korea, in 2009, an attempt was made to objectively evaluate the competence to consent for mentally ill patients by developing a valid scale to measure their competency to consent to treatment [6]. The fact that the competency to consent scale specialized for mentally impaired persons was developed might be a contradiction in that mentally impaired person’s competency to consent should be evaluated as being insufficient. Previously, without an evaluation of the competency to consent of people with mental disorders or mental disorders, it was assumed that they could not adequately consent to treatment. It also suggests that there is a high risk of making a generalization error when categorizing individuals as incompetent just because they have a mental illness without considering the severity of their symptoms. In addition, studies related to competency to consent often did not target participants using mental health services. Therefore, these studies have limitations in that they could not grasp cognitive functions important to the mentally ill living in the community.

### 1.2. Competency to Consent and Psychological Well-Being

‘Happiness’ is defined in various concepts. It is typically measured in terms of quality of life, satisfaction of life, and psychological well-being. Among them, the quality of life can be summarized as encompassing the consciousness of living a human life, feeling the happiness of everyday life, and living safely [29]. The quality of life often includes personal happiness, life satisfaction, and the quality of society that guarantees it legally and institutionally [30]. Ryff and Keyes (1995) have argued that happiness is a compilation of static functions that appear in the sub-area of psychology. They have suggested that it is well-being that provides the role of these static functions [31]. In summary, key elements of happiness include quality of life, psychological well-being, and satisfaction of life.

The mentally ill or people with mental disorders might have less opportunities to experience happiness due to social stigma or discrimination and pain caused by symptoms [5,6,14]. Various variables can affect the psychological well-being of mentally ill people. However, treatment literacy (equivalent to the ability of understanding) is related to a high quality of life associated with competency to consent of mentally ill patients [19]. Regarding the ability of application, insight has a high correlation with factors such as treatment compliance, recovery, and the life satisfaction of mentally ill patients [32,33,34]. It has been reported that the higher IQ is, the higher the self-determination and quality of life are enjoyed [35]. However, studies examining the relationship between competency to consent and psychological well-being are scarce. Since it is difficult to determine whether individual cognitive abilities affect psychological well-being, the difficulty of intervention through community programs is predicted when conducting related research.

### 1.3. Variables Related to Competency to Consent and Psychological Well-Being: Empowerment

Since the concept of empowerment is largely related to the area of choice, it is recognized as a variable that plays an important role in improving the quality of life of mentally impaired people among various factors of recovery [20,36]. Empowerment reinforces self-images such as self-esteem and self-efficacy. It enables the exercise of practical power. In addition, empowerment is a series of psychological and cognitive processes that can raise social and political consciousness. It is known to reflect the ability and decision making to autonomously control an individual’s life and environment [37]. In terms of individuals, early studies with interest in empowerment began to explain it with the concept of improving the self-competence of employees. Later, studies began to define empowerment with factors of meaning, competence, choice, and impact. Although the definition of empowerment is presented somewhat differently according to the four factors among studies, the most important part of empowerment is the area of choice that is consistently presented as the most central concept [38].

Based on the conceptual definition and previous studies, it can be inferred that empowerment, which is characterized by self-determination, is closely related to the competence to consent, which determines the right to treatment. In other words, impairment of competency to consistency may lead to impairment of empowerment. In addition, it means that empowerment can influence their psychological well-being by playing an important role in forming a positive sense of self by forming a sense of control to choose one’s life and environment. In fact, previous studies have repeatedly verified that the empowerment of mentally ill people has a positive effect on their psychological well-being. In particular, even if there are symptoms related to the recovery of mentally ill patients, research results have reported that their quality of life can be improved through social adjustment.

### 1.4. Variables Related to Treatment Consent Ability and Psychological Well-Being: Negative Emotion, Depression, Positive Emotion

The influence of emotional variables cannot be overlooked, as an individual’s internal factor can affect their psychological well-being for people with mental disorders. The result that emotional stability affects an individual’s well-being means that psychological well-being is expected to be low if an individual has negative thoughts, high guilt, high worries, and emotional instability [25]. In fact, studies have reported that people with high neuroticism of the Big Five personality traits tend to feel negative emotion with a low subjective well-being [39]. The Diagnostic and Statistical Manual of Mental Disorders (DSM)-5 includes emotional problems in the diagnostic criteria for various mental disorders or explaining sappers caused by negative moods [21]. Although positive psychology discusses the limitations of interventions with a focus on medical models, there is no doubt that over the past 100 years, psychology has accumulated vast knowledge in relation to psychopathology and contributed to symptomatic relief in people with mental illness [40].

However, positive emotions are attracting attention from researchers from a positive mental health perspective lately. The reason for this is that positive emotion plays a role in buffering harmful effects caused by negative emotion to enhance psychological well-Being [41]. Explaining this effect is as follows. When the ratio between positive and negative emotion exceeds 3:1, it is said that the level of mental health can reach a state of flooding [42]. In other words, positivity can increase resilience and offset chronic mood problems known to be main symptoms of mentally ill people, thus making them feel happy. This can be the basis for assuming that positive emotion can effectively deal with psychopathology in a positive psychotherapy approach that attempts to improve strength and prevent a recurrence [43]. Therefore, it can be seen that a study on the feeling of happiness, which places importance on the positive emotion of mentally ill patients, has important implications not only in terms of recovery of mentally ill patients but also in the therapeutic aspect of psychopathology.

### 1.5. Research Purpose

By understanding the competency to consent to treatment of mentally ill and mentally impaired people, this study attempts to determine whether they have problems in self-determination and sub-factors of individual cognitive ability. This study also aims to determine which factors might have a greater relevance to psychological well-being. To this end, for psychiatric outpatients, factors that can affect psychological well-being in addition to their competency to consent to treatment level are actually identified.

## 2. Materials and Methods

### 2.1. Research Subject

This research was conducted with 196 patients who agreed to participate in this study among patients undergoing outpatient treatments at the Department of Mental Health in Gyeongsangnam-do and Jeonranam-do areas, Korea. After excluding data of five patients who responded insincerely (e.g., insincere responses are uniformly answered with one number on one or more scales despite the fact that there are inverse questions), data of 191 patients were finally analyzed. All study participants were diagnosed according to the diagnostic criteria of DSM-5, including those with depressive disorder (33.2%), panic disorder (14.8%), schizophrenia (8.2%), sleep disorder (14.8%), bipolar disorder (4.3%), anxiety disorder (13.3%), post-traumatic stress disorder (2.0%), obsessive-compulsive disorder (1.2%), neurocognitive disorder (0.4%), attention deficit hyperactivity disorder (ADHD) (2.7%), substance use disorder (1.6%), and others (3.5%).There were 63 males and 128 females. Their average age was 36.49 (SD: 11.13) years. There were 155 (81.2%) participants who had no previous hospitalization experience and 36 (18.8%) who had prior hospitalization experience. Regarding education level, there was 1 (0.5%) dropout from elementary school, 5 (2.6%) elementary school graduates, 8 (4.2%) junior high school graduates, 90 (47.1%) high school graduates, 15 (7.9%) students in college, 67 (35.1%) college graduates, 2 (1.0%) graduate students, and 2 (1.0%) graduates from graduate school.

### 2.2. Measuring Tool

#### 2.2.1. Competency to Consent to Treatment

To evaluate the competency to consent to treatment of psychiatric outpatients, the Korean-style competency to consent to treatment evaluation tool developed and validated by Seo et al. [6] was used. This evaluation tool was conducted in the following manner as a structured interview. A script describing general symptoms that a mentally ill person might experience (such as decreased concentration and motivation, insomnia, anxiety, and psychotic symptoms, the need and merits of drug treatment, and the problems that might occur when not treated) was read. Then, they answered questions of the script for evaluating their ability to express, understand, apply, and reason. There were a total of 22 questions. Each question was scored as follows: 0–2 points (3 questions) and 0–1 points (4 questions) for the ability of understanding, 0–2 points (2 questions) for the ability of application, 0–1 point (4 questions) for the ability to express, and 0–2 points (2 questions) and 0–1 point (4 questions) for the ability of reason. There are a few points to note with the above tool. The four sub-components of the above-mentioned ability to consent to treatment require an independent evaluation and the completeness of one ability should not be considered sufficient for another. The cut-off for each sub-factor is as follows: overall score, 18.5 points; ability of understanding, 4.5 points; ability of application, 8.5 points; ability to express, 0.5 points; and the ability of reasoning, 3.5 points. In the study of Seo et al. [6], Cronbach’s α value for each variable was as follows: 0.83 for overall scale, 0.70 for the ability of understanding, 0.75 for the ability of application, 0.56 for the ability to express, and 0.71 for the ability of reasoning. In the present study, Cronbach’s α values were as follows: 0.82 for overall scale, 0.70 for the ability of understanding, 0.63 for the ability of application, 0.65 for the ability to express, and 0.74 for the ability of reasoning.

#### 2.2.2. Psychological Well-Being Scale (PWBS)

The psychological well-being scale (PWBS) used in this study was developed by Ryff [44] and validated by Kim et al. [45]. This scale was composed of sub-factors such as self-acceptance, positive interpersonal relationships, autonomy, control over the environment, the purpose of life, and personal growth. The rating was based on a 5-point Likert scale of 46 questions (from 1 point for ‘not at all’ to 5 points for ‘very much’). In the study of Kim et al. [45], Cronbach’s α values for sub-factors of this scale were as follows: 0.76 for self-acceptance, 0.72 for positive interpersonal relationships, 0.73 for life purpose, 0.70 for personal growth, 0.69 for autonomy, and 0.66 for environmental control. Cronbach’s α values in this study were as follows: 0.88 for self-acceptance, 0.82 for positive interpersonal relationships, 0.82 for life purpose, 0.73 for personal growth, 0.76 for autonomy, and 0.79 for environmental control.

#### 2.2.3. Positive Affect and Negative Affect Schedule (PANAS)

The positive affect and negative affect schedule (PANAS) developed by Lee et al. [46] was used to determine the overall level of positive and negative emotion of psychiatric outpatients in the present study. This scale consisted of 10 questions to measure positive emotion and 10 questions to measure negative emotion. Each question was rated on a 7-point Likert scale (from 0 point for ‘don’t feel at all’ to 6 points for ‘feel very much’). In the present study, its Cronbach’s α value was found to be 0.75.

#### 2.2.4. The Korean Version of the Patient Health Questionnarie-9 (PHQ-9)

The Korean version of the depression screening tool (Patient Health Questionnarie-9; PHQ-9) developed by An et al. [47] was used to determine the degree of depression in psychiatric outpatients of the present study. This scale was composed of a single factor with 7 questions. Each question was rated on a 4-point Likert scale (from 1 point for ‘No at all’ to 4 points for ‘Almost every day’). In the study of An et al. [47], Cronbach’s α of this scale was 0.95. In the present study, its Cronbach’s α value was 0.89.

#### 2.2.5. Empowerment

To determine the degree of empowerment, the empowerment scale developed by Rogers et al. [23] and validated by Oh [48] was used. This scale was composed of sub-factors of self-efficacy, optimistic perspective, fair anger, self-esteem, helplessness–strength, community activity, and autonomy with a total of 23 questions Each question was rated on a 5-point Likert scale (from 0 point for ‘always not’ to 5 points for ‘always is’). In Oh’s [48] study, Cronbach’s α of this scale was calculated as 0.85. In the present study, its Cronbach’s α was found to be 0.92.

### 2.3. Procedure

This study was conducted after obtaining approval from the Institutional Bioethics Committee (IRB) of Gyeongsang National University (GIRB-A20-W-0013). The purpose and procedure of this study were explained to participants before conducting the questionnaire and interview. Detailed information on guaranteeing anonymity, not to use the data for purposes other than research, and when to keep and discard the data were provided. After receiving consent from participants to participate in the study, interviews and questionnaires were conducted.

### 2.4. Analysis Method

The purpose of this study was to determine the competency to consent to treatment of psychiatric outpatients and to examine the role of empowerment and emotion in the relationship between competency to consent and psychological well-being scale (PWBS). The analysis was performed using SPSS 25.0 version and SPSS Process Macro version 3.1. Specifically, the internal consistency coefficient (Cronbach’s α) of each measurement tool was confirmed. The relationship between competence to consent and psychological well-being scale (PWBS) was determined based on correlation analysis. For verified results, a dual mediation effect was verified by utilizing variables related to empowerment and emotions in order to provide a more expanded understanding.

## 3. Results

### 3.1. Psychiatric Outpatient’s Competency to Consent to Treatment

The results were based on cut-off points for the overall score and sub-factor scores for competency to consent to treatment in a study by Seo et al. [6]. Accordingly, as a result of checking the scores of the participants in this study, their average competency to consent, ability of understanding, ability of application, ability to express, and ability of reasoning were 30.69, 7.90, 12.00, 3.90, and 6.90 points, respectively, showing good levels overall.

As a result of qualitative evaluation of the presence or absence of competency to consent to treatment, the number of patients who scored less than the cut-off point was as follows: 7 for total score, 22 for the ability of understanding, 13 for the ability of application, 1 for the ability to express, and 9 for the ability of reason. Regarding competency to consent according to hospitalization, patients with hospitalization experience scored lower than those without hospitalization experience for the following variables: overall competency to consent (t = 3.33, *p* < 0.01), understanding (t = 3.19, *p* < 0.01), application (t = 3.40, *p* < 0.001), and reasoning (t = 2.41, *p* < 0.01).

The difference in competency to consent according to gender was not significant for all sub-factors, including the overall score. The difference in competency to consent according to age group was verified to be significant for the ability of understanding among sub-factors (F = 4.31, *p* < 0.05). As a result of post hoc analysis, it was found that those who were in their 20 s, 30 s, and 40 s had higher ability of understanding than those who were in their 50 s or older.

### 3.2. Correlation Analysis

Results of correlation analysis that could be meaningfully interpreted among verified correlation results are as follows. The overall score of competency to consent was not significantly correlated with the psychological well-being scale (PWBS). However, a positive correlation between ability of application and psychological well-being scale (PWBS) was confirmed (r = 0.20, *p* < 0.01). For correlation analysis of empowerment, only the ability of application showed a significant positive correlation with the empowerment of mentally impaired (r = 0.21, *p* < 0.01). Empowerment showed correlations with depression (r = 0.58, *p* < 0.00), positive emotion (r = 0.64, *p* < 0.00), and negative emotion (r = −0.58, *p* < 0.00). The psychological well-being scale (PWBS) also showed significant correlations with depression (PHQ-9) (r = −0.66, *p* < 0.001), positive emotion (r = 0.73, *p* < 0.001), and negative emotion (r = −0.65, *p* < 0.001).

These results confirmed that the ability of application among sub-factors of competency to consent to treatment was related to the psychological well-being scale (PWBS) and empowerment of psychiatric outpatients. In addition, it was verified that there were statistically significant correlations of empowerment with depression, positive emotion, negative emotion, and psychological well-being scale (PWBS). Subsequent analysis was conducted to determine what roles empowerment, depression, positive emotion, and negative emotion might play in the relationship between the ability of application and psychological well-being scale (PWBS). Results of the correlation coefficient of the whole scale are presented in Table 1.

### 3.3. Analysis of the Dual Mediation Effect of Empowerment and Negative Emotion

In this study, the dual mediation effect of empowerment and emotion was firstly verified in the relationship between ability of application and psychological well-being score (PWBS). This is because emotional problems (negative emotions) are common symptoms of people with a mental illness. Detailed results are as follows. The ability of application was found to have a statistically significant effect on psychological well-being score (PWBS) (β = 0.061, *p* < 0.001) and empowerment (β = 0.071, *p* < 0.001). When negative emotion was adopted as a dependent variable and the effectiveness of application and empowerment were examined, the ability of application did not have a significant effect (β = 0.041, *p* > 0.05), whereas empowerment had a statistically significant effect (β = −1.239, *p* < 0.001). Effects of the ability of application, empowerment and negative emotion on psychological well-being score (PWBS) as a dependent variable are as follows. The ability of application was not statistically significant (β = 0.010, *p* > 0.05). However, empowerment (β = 0.642, *p* < 0.001) and negative emotion (β = −0.095, *p* > 0.001) were found to have statistically significant effects. These results indicate that the relationship between the ability of application and psychological well-being score (PWBS) is mediated by both empowerment and negative emotion. These results are presented in Table 2 and Figure 1.

As a result of verifying the indirect effects of empowerment and negative emotion with bootstrap, the value of the overall mediation effect was 0.050. If 0 was not included between the lower confidence interval (0.009) and the upper confidence interval (0.090) for the 95% confidence interval, it indicated statistical significance. When verifying the simple mediation effect, the value of the indirect effect for the route through empowerment was measured to be 0.046. There was no zero between the lower confidence interval (0.013) and the upper confidence interval (0.080) of the 95% confidence interval, indicating statistical significance. However, the path through negative emotion had an indirect effect value of -0.004, which was not statistically significant because 0 was included between the lower confidence interval (−0.014) and the upper confidence interval (0.005) of the 95% confidence interval. Finally, the double-mediated path through negative emotion after empowerment had an indirect effect size of 0.009. The dual mediation effect was found to be statistically significant because 0 was not included between the lower confidence interval (0.003) and the upper confidence interval (0.016) at 95% of the confidence interval. Results are summarized in Table 3.

### 3.4. Analysis of Dual Mediation Effect of Empowerment and Depression

In the relationship between the ability of application and the psychological well-being score (PWBS), which was verified in the previous analysis, the dual mediation effect of the symptom-centered variable that further developed one step more from empowerment and negative emotion was verified. For verification, model 6 of SPSS Process Macro was used with bootstrapping set to be 5000.

First, a dual mediation effect of empowerment and depression in the relationship between the ability of application and psychological well-being scale (PWBS) was confirmed. Specifically, the ability of application was found to have a significant effect on the psychological well-being scale (PWBS) (β = 2.595, *p* < 0.001) and empowerment (β = 0.070, *p* < 0.001). When depression was assumed as a dependent variable and effects of application and empowerment were examined, the ability of application did not have a statistically significant effect (β = −0.006, *p* > 0.05), whereas empowerment did have a statistically significant effect (β = 0.647, *p* < 0.001). After examining effects of ability of application, empowerment, and depression (PHQ−9) on the psychological well-being scale (PWBS) as a dependent variable, the effect of ability of application was not statistically significant (β = 0.004, *p* > 0.05), whereas effects of empowerment (β = 0.639, *p* < 0.001) and depression (PHQ−9) (β = −0.191, *p* < 0.001) were statistically significant. These results indicate that the relationship between the ability of application and psychological well-being scale (PWBS) is dual mediated by empowerment and depression (PHQ-9). These results are presented in Table 4 and Figure 2.

As a result of verifying indirect effects of empowerment and depression (PHQ-9) through bootstrap, the total mediation effect value was 0.055. In the 95% confidence interval, zero was not included between the lower confidence interval (0.015) and upper confidence interval (0.096), indicating a statistically significant effect. During simple mediation effect verification, the indirect effect value of the route via empowerment was 0.045. There was no zero between the lower confidence interval (0.014) and the upper confidence interval (0.078) of the 95% confidence interval, indicating statistical significance. However, the path through depression (PHQ-9) had an indirect effect value of 0.001. There was zero between the lower confidence interval (0.003) and the upper confidence interval (0.016) of the 95% confidence interval, indicating no statistical significance.

Finally, for the path of depression (PHQ-9) after empowerment, the value of the indirect effect was 0.009. The dual mediation effect was statistically significant because zero was not included between the lower confidence interval (0.003) and the upper confidence interval (0.016) at 95% of the confidence interval. These results are shown in Table 5.

### 3.5. Dual Mediation Effect of Empowerment and Positive Emotion

Finally, the dual mediation effect of empowerment and positive emotion was verified in the relationship between the ability of application and psychological well-being scale (PWBS). This is done because the absence of a negative emotion does not guarantee a positivity. In addition, a positive emotion might have another effect on the psychological well-being. Specifically, the ability of application had a statistically significant positive effect on the psychological well-being scale (PWBS) (β = 0.061, *p* < 0.001) and empowerment (β = 0.071, *p* < 0.001). When a positive emotion was adopted as a dependent variable for examining effects of application and empowerment, the ability of application did not have a statistically significant effect (β = −0. 032, *p* > 0.05). However, empowerment had a statistically significant effect (β = 1.351, *p* < 0.001). Effects of the ability of application, empowerment, and positive emotion on the psychological well-being scale (PWBS) as a dependent variable were also examined. Although the ability of application was not statistically significant (β = 0.010, *p* > 0.05), empowerment (β = 0.587, *p* < 0.001) and positive emotion (β = 0.129, *p* < 0.001) were found to have a statistically significant effect, respectively. These results indicate that the relationship between the ability of application and psychological well-being scale (PWBS) is dual mediated by empowerment and positive emotion. These results are presented in Table 6 and Figure 3.

As a result of verifying the indirect effects of empowerment and positive emotion with bootstrap analysis, the total mediation effect value was 0.0502. In the 95% confidence interval, zero was not included between the lower confidence interval (0.009) and the upper confidence interval (0.088), indicating statistical significance. As for a simple mediation effect, the path through empowerment had an indirect effect value of 0.042. There was no zero between the lower confidence interval (0.014) and the upper confidence interval (0.078) of the 95% confidence interval, indicating significance. The value of the indirect effect of the route through a positive emotion was −0.004. It was found that the 95% confidence interval contained zero between the lower confidence interval (−0.017) and the upper confidence interval (0.007), indicating no statistical significance. Lastly, the value of the indirect effect was 0.0124 for the dual mediation effect through positive emotion after empowerment. The dual mediation effect was statistically significant because zero was not included between the lower confidence interval (0.004) and upper confidence interval (0.024) at 95% of the confidence interval. These results are presented in Table 7.

## 4. Discussion

### 4.1. Results and Implications

In this study, the competency to consent of psychiatric outpatients was investigated. In addition, effects of empowerment and depression on the relationship between competency to consent and psychological well-being were examined. The main results of this study are as shown below:

#### 4.1.1. Competency to Consent of Psychiatric Outpatient

Most study participants scored above the cut-off level. Thus, the overall competency to consent of these psychiatric outpatients was good. In addition, it was confirmed that patients without hospitalization experience scored lower than those with hospitalization experience. Patients with hospitalization experience have received training related to symptoms during hospitalization and undergone the process of accepting their symptoms. Through this process, their ability of understanding and ability of application can be improved. Thus, the likelihood that their competency to consent for treatment has improved could be increased. In addition, they showed a score of 22 for their ability of understanding, 13 for their ability of application, and 9 for their ability of reasoning, which were below the cut-off level for each sub-scale. Nevertheless, only seven people scored below the cut-off level of the overall scale. This partly confirmed results of previous studies showing the risk of seeing each cognitive ability value as incompetent and the importance of evaluating each cognitive ability individually [6,18]. In particular, according to previous studies, the most basic sub-variable that judges that one has no competency to consent is the ability to express [6]. For this ability, no psychiatric outpatient scored below the cut-off level. Thus, it is appropriate to assume that psychiatric outpatients possess a basic competency to consent. The ability to understand and recognize information about symptoms and treatment methods is related to knowledge that can be sufficiently supplemented through education. The ability of reasoning to logically determine treatment rejection and acceptance is also likely to be improved through the provision of accurate information [49]. Although it might be a bit slower than reinforcing knowledge to improve their ability of application related to knowledge, there is sufficient evidence that it is possible to improve their insight through constant biopsychosocial intervention and education [50,51].

Based on these results, it is necessary to examine why mentally ill people have a negative view of them despite having sufficient cognitive ability to make decisions about their treatment. This was because this perspective was formed by the social representation shared by our social community. Social representation is the way we understand what we know and how we share what we know with others [52]. The cognitive category of social representation includes a group’s knowledge, values, and information about an object. It is also an attitude that reflects a group’s positive and negative emotional evaluations for a specific object [53]. In addition, since attitude is a product of awareness of a certain situation [54], social representation can be said to be an appropriate medium when grasping the attitude toward a mentally ill person. A negative social representation of mental illness will increase the likelihood of causing human rights problems in various areas of society. In particular, in an information society, mass media plays a very important role in the formation of the general public’s attitude toward mental illness [55]. Therefore, objective information on mental disorders, not simply listing prohibited actions, should be delivered, and educational content that can enhance our understanding of mental disorders is needed for human rights education.

#### 4.1.2. Competency to Consent Is Important to Users of Mental Health Services: Ability of Application

Among the four sub-variables of competency to consent, only the ability of application was found to be correlated with PWBS and empowerment. The ability of application refers to the ability to assign information on mental disorders to oneself. It is known to be the most relevant to a patient’s insight [18]. Previous studies have reported that the ability of understanding and the ability of reasoning are factors predicting the quality of life of patients. However, in the case of psychiatric outpatients, even if they can evaluate the validity based on logic with knowledge of the disability, if they cannot be applied to their own lives, it means that it has nothing to do with the aspect of enhancing an individual’s empowerment and quality of life. In previous studies, insight refers to a comprehensive process that includes understanding intellectual information and additionally maintaining motivation for treatment, while the ability of application refers to a microscopic concept that recognizes that one has a disability who can apply the intellectual information as understood by oneself [18,56]. This means that for psychiatric outpatients, accommodating their illness and coping with their problem-solving situations in the community in terms of improving environmental control and psychological well-being are key factors. To this end, mental health service organizations in the community can focus on individual symptoms. In addition, psychological interventions that can use appropriate social skills in various problem-solving scenes can be emphasized.

#### 4.1.3. The Mediating Effect of Empowerment on the Ability of Application and Psychological Well-Being

The relationship between the ability of application and psychological well-being was found to be fully mediated by empowerment. In the empowerment system, even if an individual may suffer or become oppressed in a system or relationship in which power operates, the individual is a dignified being with the will and ability to overcome this and create a better environment [37]. The concept of empowerment will be a key element that must be reinforced for outpatients for whom it is natural that living in the community is possible. However, practical incompetence should not be premised to be recognized for their human rights. When the ability of application is increased, the empowerment for an individual with a mental illness is improved. Social exclusion caused by an individual avoiding the formation of relationships on their own can be overcome. In addition, even individuals who lack the ability of application are likely to show increased psychological well-being if there are system-centered interventions that can improve empowerment. In the present society, the mental health paradigm that can lead to daily return is emphasized. Research results have proven that empowerment is a very important factor.

However, reality sometimes operates differently from an ideal concept. Previous studies have shown inconsistent results for facilities and institutions that have introduced an empowerment system. Even in studies conducted in Korea, results are not different from previous studies. Considering the domestic situation, it was confirmed that differences between institution settings appeared in studies that looked at the level of empowerment and quality of life of various institutions that provided mental health services in Korea [57]. In the present study, the researcher insisted on a transitional effect of the empowerment welfare system and emphasized the concept of substantial empowerment. In addition, this study discussed the risk that if the community could not provide substantial empowerment to the mentally ill, empowerment would have only free structure in an emotionally and psychologically closed form that might not be significantly different from admission facilities or inpatient wards.

In a sociocultural atmosphere where the social stigma of mentally ill patients is severe, it is difficult for them to accept their symptoms. It is also difficult to accurately recognize mental diseases whose symptoms are invisible, unlike physical diseases [6,7,9,11]. Improving insight is not an easy task, as it is difficult for people with mental illness to change the schema within them [50]. In this situation, the role of empowerment should be emphasized. It is important not only to introduce concepts and practices but also to establish a sound mental health service system that can achieve the purpose of empowerment through concrete changes at both psychological and organizational levels.

#### 4.1.4. The Double Mediating Effect of Empowerment and Emotion on the Relationship between the Ability of Application and Psychological Well-Being

The emotional pain of a mentally ill person is already understandable even at the level of common sense. Since psychiatric outpatients are members of the community system, it is appropriate to examine an individual’s empowerment and emotional state together. In addition, in the relationship between the ability of application and psychological well-being, even if empowerment is completely mediated, it is also important to understand the influence of a third variable. As a result, it was found that the ability of application affected empowerment, lowering negative emotion and increasing positive emotion, thereby improving psychological well-being. If the ability of application is high, social participation will increase, thus lowering negative emotion and depression and improving positive emotion, resulting in a higher psychological well-being emotion. This confirms that the influence of an individual’s control power is important in emotional experience [58]. Assuming the incompetence of people with mental illness is a response that ignores their control or decision-making power; this response is a macroscopic problem of human rights that violates their basic rights. It can lead to problems from a microscopic point of view of increasing negative emotions and weakening positive emotions, which can worsen individual mental health.

The effect of emotion means that the need for a medical model of clinical psychology that is currently being criticized must be combined with the result of enhancing strength from a positive psychological perspective [40]. Thus, mental health workers should try their best in their natural roles so that patients themselves have motivation for treatment and adaptation, which can lead to increased individual happiness. This confirms that mutual cooperation between various institutions, including psychiatric outpatient clinics, is an important factor for minorities to pursue happiness. The results of this study showed that the ability of application, improvement of empowerment, and emotional reduction were all important. However, the direction in which emotions directly affect a happy life can be a key factor in preventing symptoms. The way people with mental disorders are accommodated within the social system plays an important role in improving their control. Results of this study suggests that it is necessary to supplement the system and secure the rights and interests of minorities through national efforts.

### 4.2. Limitations of the Study

Limitations of this study are as follows. First, in this study, the selection of subjects was somewhat limited. This was because their voluntary consent was required when selecting a subject. Thus, all participants in the current study had competency to consent to participate in the study. Therefore, patients with serious psychotic problems or neuropsychological problems could not be included in this study. In the follow-up study, in order to obtain data for patients with severe symptoms, it will be necessary to evaluate their competence to consent and analyze them with the approval of their legal representatives.

Second, human rights problems, such as violation of the right to self-determination during treatment and hospitalization, can also occur in patients who visit mental health-related institutions for the first time. However, the competency to consent evaluation tool used in the current study could not be used for first-time patients because there were questions that could not be answered without the patient’s actual diagnosis. To minimize various human rights issues related to competency to consent, it is necessary to develop a competency to consent evaluation tool for first-time patients.

Third, in order to evaluate the competence to consent, time and economic costs are considerable. Cooperation from related organizations is required. Therefore, subjects of this study were limited to psychiatric outpatients in Gyeongsangnam-do and the Jeonranam-do regions. Accordingly, it is difficult to generalize the results of this study to all domestic psychiatric outpatients. To overcome this limitation, it is necessary to conduct national-level studies in the future.

Fourth, in this study, it was difficult to understand the influence of sub-variables other than the ability of application because there were not enough previous studies on competency to consent to sufficiently explain a specific phenomenon or individual characteristics and attributes. To reliably predict the psychological well-being of psychiatric patients through competency to consent, follow-up studies on various variables other than the one measured in this study will be needed. In addition, most psychiatric outpatients scored above the cut-off level on the competency to consent scale. Therefore, it is desirable to evaluate their competency to consent with a method other than the method used in the present study. Using the results found in this study as basic data, it is necessary to re-establish the theory on the concept of competency to consent suitable for psychiatric outpatients so that specialized scales for them could be developed. Through this, it will be possible to grasp their competence to consent, taking into account the difference according to the severity of their symptoms.

## 5. Conclusions

In summary, it is very important to feel a sense of social belonging for outpatients who share social activities with family, friends, and co-workers not in hospitalization facilities. In this way, the increased sense of control can go beyond just bad or good feelings to relieve symptoms such as depression. As a result, it is more likely to maintain their functional and adaptive behavior. What people with mental illness need in order to belong to the society are of course important to their cognitive abilities, such as competency to consent. However, what should be more emphasized is to change the social atmosphere in which people with mental illness are perceived as “incompetent” regardless of the severity of their symptoms. It is necessary to correct the perception that people with mental illness are dangerous and uncontrollable because they are receiving medication in psychiatry. It is important to establish and operate various positive institutions so that they can participate in therapeutic and preventive interventions while controlling symptoms in the community. In addition, not only social and systematic efforts but also individuals experiencing mental illness must maintain an effort to accurately grasp their psychological state. Through this, it should be noted that participating in education that improves their ability of application is a way to secure one’s right to pursue happiness. It is desirable for the community to co-exist through harmony. This needs to be achieved through efforts of both individuals and the society.

## Figures and Tables

**Figure 1 ijerph-18-08170-f001:**
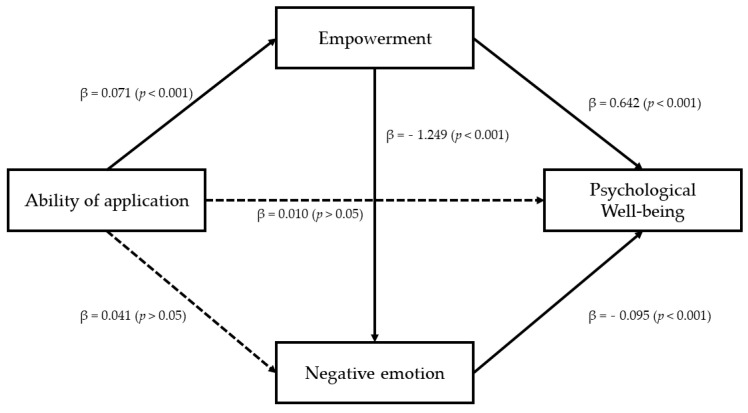
The result of verification of the double mediating effect of empowerment and negative emotion.

**Figure 2 ijerph-18-08170-f002:**
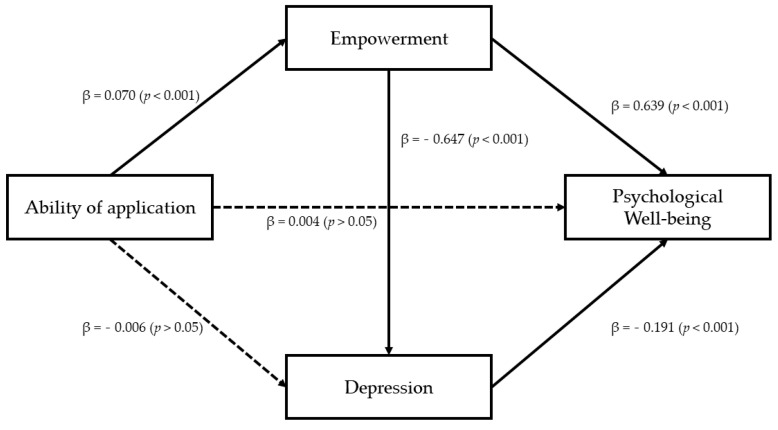
The result of verification of the double-mediating effect of empowerment and depression.

**Figure 3 ijerph-18-08170-f003:**
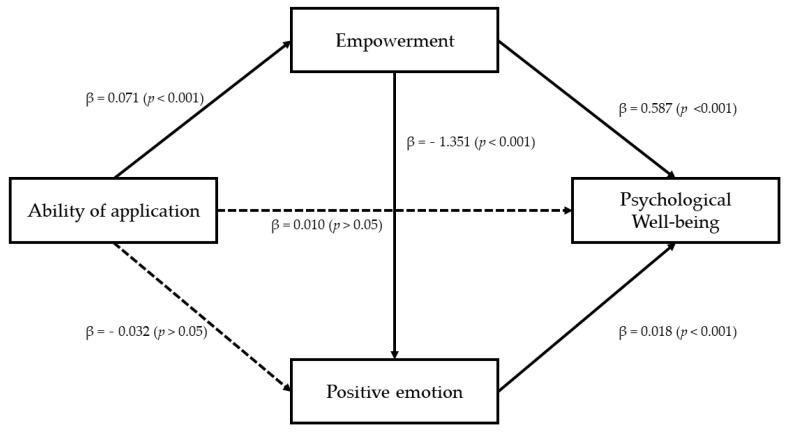
The result of verification of the double mediating effect of empowerment and positive emotion.

**Table 1 ijerph-18-08170-t001:** Correlation among variables (*n* = 191).

	1	2	3	4	5	6	7	8	9
Competency to consent to treatment	-								
Ability to understand	0.793 ***	-							
Ability to apply	0.837 ***	0.439 ***	-						
Ability to express	0.316 ***	0.091	0.255 ***	-					
Ability to reason	0.801 ***	0.431 ***	0.622 ***	0.246 ***	-				
Psychological well-being	0.114	−0.031	0.204 **	0.070	0.121 *	-			
Negative emotion	0.014	0.082	−0.068	0.021	0.004	0.601 ***	-		
Positive emotion	−0.010	−0.178 **	0.092	0.108	0.078	−0.442 ***	−0.497 ***	-	
Depression	−0.108	−0.022	−0.141 *	−0.045	−0.113	0.534 ***	0.710 ***	−0.597 ***	-
Empowerment	0.152 *	0.019	0.213 **	0.093	0.149 *	−0.493 ***	−0.582 ***	0.639 ***	−0.576 ***

1 = competency to consent to treatment, 2 = ability of understanding, 3 = ability of application, 4 = ability to express, 5 = ability of reasoning, 6 = psychological well-being, 7 = negative emotion, 8 = positive emotion, 9 = depression. * *p* < 0.05, ** *p* < 0.01, *** *p* < 0.001.

**Table 2 ijerph-18-08170-t002:** Mediation effect of empowerment and negative emotion.

Step	DV	IV	β	SE	t	Lower CI	Upper CI	R^2^	F
1	Psychological well-being	Ability of application	0.061	0.021	2.858	0.019	0.102	0.04	8.17 ***
2	Empowerment	Ability of application	0.071	0.024	2.984 ***	0.024	0.119	0.05	8.91 ***
3	Negative emotion	Ability of application	0.041	0.043	0.957	−0.044	0.126	0.34	48.48 ***
		Empowerment	−1.249	0.123	−9.79 ***	−1.501	−0.997		
4	Psychological well-being	Ability of application	0.010	0.010	0.973	−0.010	0.031	0.78	224.39 ***
		Empowerment	0.642	0.038	16.935 ***	0.567	0.717		
		Negative emotion	−0.095	0.018	−5.380 ***	−0.130	−0.060		

DV: Dependent variable, IV: Independent variable, β: Standardized regression weights, SE: Standard error, t: t-value, CI: Confidence interval, R^2^: r-square, F: f-value *** *p* < 0.001.

**Table 3 ijerph-18-08170-t003:** Verification of indirect effect of empowerment and negative emotion.

Path				Effect	SE	95% CI	
						Lower CI	Upper CI
Ability of application	Empowerment		Psychological well-being	0.046	0.017	0.013	0.080
Ability of application	Negative emotion		Psychological well-being	−0.004	0.005	−0.014	0.005
Ability of application	Empowerment	Negative emotion	Psychological well-being	0.009	0.003	0.003	0.016
Total indirect effect				0.050	0.021	0.009	0.090

SE: Standard error, CI: Confidence interval.

**Table 4 ijerph-18-08170-t004:** Mediation effect of empowerment and depression.

Step	DV	IV	β	SE	t	Lower CI	Upper CI	R^2^	F
1	Psychological well-being	Ability of application	0.059	0.021	2.790 ***	0.017	0.101	0.04	7.79 ***
2	Empowerment	Ability of application	0.070	0.024	2.958 ***	0.023	0.117	0.04	8.75 ***
3	Depression	Ability of application	−0.006	0.023	−0.264	−0.052	0.039	0.33	46.81 ***
		Empowerment	−0.647	0.069	−9.401 ***	−0.782	−0.511		
4	Psychological well-being	Ability of application	0.004	0.204	0.400	−0.016	0.024	0.79	232.71 ***
		Empowerment	0.639	0.037	17.321 ***	0.566	0.712		
		Depression	−0.191	0.032	−5.921 ***	−0.255	−0.127		

DV: Dependent variable, IV: Independent variable, β: Standardized regression weights, SE: Standard error, t: t-value, CI: Confidence interval, R^2^: r-square, F: f-value. *** *p* < 0.001.

**Table 5 ijerph-18-08170-t005:** Verification of indirect effect of empowerment and depression.

Path				Effect	SE	95% CI	
						Lower CI	Upper CI
Ability of application	Empowerment		Psychological well-being	0.045	0.016	0.014	0.078
Ability of application	Depression		Psychological well-being	0.001	0.005	−0.009	0.011
Ability of application	Empowerment	Depression	Psychological well-being	0.009	0.004	0.003	0.016
Total indirect effect				0.055	0.021	0.015	0.096

SE: Standard error, CI: Confidence interval.

**Table 6 ijerph-18-08170-t006:** Mediation effect of empowerment and positive emotion.

Step	DV	IV	β	SE	t	Lower CI	Upper CI	R^2^	F
1	Psychological well-being	Ability of application	0.061	0.021	2.858	0.019	0.102	0.04	8.17 ***
2	Empowerment	Ability of application	0.071	0.024	2.984 ***	0.024	0.119	0.05	8.91 ***
3	Positive emotion	Ability of application	−0.032	0.040	−0.194	−0.111	0.047	0.41	65.21 ***
		Empowerment	1.351	0.120	11.300 ***	1.115	1.587		
4	Psychological well-being	Ability of application	0.010	0.010	1.044	−0.009	0.905	0.80	254.40 ***
		Empowerment	0.587	0.038	15.416 ***	0.512	0.662		
		Positive emotion	0.129	0.018	7.174 ***	0.093	0.164		

DV: Dependent variable, IV: Independent variable, β: Standardized regression weights, SE: Standard error, t: t-value, CI: Confidence interval, R^2^: r-square, F: f-value. *** *p* < 0.001.

**Table 7 ijerph-18-08170-t007:** Verification of indirect effect of empowerment and positive emotion.

Path				Effect	SE	95% CI	
						**Lower CI**	**Upper CI**
Ability of application	Empowerment		Psychological well-being	0.042	0.015	0.013	0.072
Ability of application	Positive emotion		Psychological well-being	−0.004	0.006	−0.017	0.007
Ability of application	Empowerment	Positive emotion	Psychological well-being	0.012	0.005	0.004	0.024
Total indirect effect				0.050	0.020	0.009	0.088

SE: Standard error, CI: Confidence interval.

## Data Availability

Restrictions apply to the availability of these data. Data was obtained.

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
