# Peer review of "Relationship between Competency to Consent to Treatment and Psychological Well-Being: Mediating Effect of Empowerment and Emotion"

_ijerph, 2021, doi:10.3390/ijerph18158170_

Round 1

Reviewer 1 Report

In general I consider that it deals with a very interesting topic, and does not go so far as to give value judgments on such a sensitive subject. The language of the article is generally speaking appropriate and understandable.

Abtsract: please indicate what type of analysis you performed.

Introduction: line 55 and 59 are the same. Reread your document, I feel it is repetitive from time to time. I understand that technical concepts do not have many synonyms, but remember that the intention is to make the document attractive to the reader (you could even make it shorter). Please do not forget to mention some statistical data on the prevalence of mental illness in Korea (and maybe the region). Lines 99-105 do not belong to the introduction section (maybe at the participants section). Sometimes I feel you jump from one topic to another; it disrupts the fluidity of the text. From my perspective, finding the research objective after five pages of introduction is somehow difficult, especially because the concepts come one after another without having a clear relationship between them. Additionally, I suggest the use of subheadings. Lines 233-237, seems to fit better in the procedure section.

2.1. Research subject: how did you determine that 5 people responded insincerely?

Competency to consent to treatment: why do you use two different types of hyphens when talking about the score? (- y ~)?

Lines 316-318 repeat the objective (there is no need).

Procedure and Psychiatric outpatient's competency to consent to treatment: I understand your objective was to study the competency to consent of psychiatric outpatients and study the effects of empowerment and depression on the relationship between competency to consent and psychological well-being. Nevertheless, most of the time, it seems like a psychometric validation, even a comparison of your psychometric properties with the ones found in other studies. I consider that it is unnecessary to fill with data and psychometric values, a text that is already heavy and full of analysis and variables. This is not to say that you do not report your alphas, but that you find a better way to express that the instruments are valid (perhaps a table could help summarize everything you found in this regard). I would like to see the descriptives of the "Competency to consent to treatment" tool, including the floor and ceiling cases (below and above cutoff), taking into account age and hospitalization experience (the results you report as significant). Analysis of the dual mediation effect: What do you think about the use of graphic resources to show the relationships found?

Discussion and conclusions: Authors should focus their discussion on the findings of their work. At times I appreciate that these sections seek to study the social representations of patients rather than delving into their objective "the competency to consent of psychiatric outpatients" (the conclusions do not contain even a mention of this element). For instance, after repeatedly reading lines 498-513, I do not quite understand the authors' point of view regarding the results.

Finally, the Data Availability Statement seems incomplete.

Author Response

Dear. Reviewer 1.

Thank you for your review.
Unfortunately, in the case of the details of the review evaluation, it was uploaded as an attachment due to the cunning and a lot of content. Please confirm.

I would like to express my gratitude to the attentive and meticulous couple.

Reviewer 2 Report

Thank you for providing a chance to review your manuscript. Overall, the paper is well-written, but minor revisions are still needed. Overall, this is an interesting study and provides a better understanding of the relationship between competency to consent to treatment and psychological well-being in Psychiatric patient. In the age of human rights, this research makes perfect sense.

1.Introduction section.

The researcher's theme has great significance, but the author fails to show it, especially in the context of Eastern culture.

Kleinman points out that mentally ill people are stigmatized because they deviate from the cultural conventions of a given society's definition of what is acceptable in terms of appearance and behavior, and are instead defined as ugly, scary, alien and inhuman. Based on comparative studies, Kelemen et al. revealed that the sociogenetics of stigma varies in different societies: In the Confucian society, psychiatric patients and their families were stigmatized main reason is the destruction of the social relation network, face and disgrace concept in understanding the phenomenon of stigma of mental illness is of great significance in the society have long-term stigmatized, and even used as a curse, this kind of "demonizing" the behavior of mental illness, leading people to stay at a respectful distance from sb in patients with mental illness. In fact, mental illness is the same as other diseases, just different parts of the disease, after treatment, patients can mostly return to normal work and life. The psychological burden has become the psychological obstacle that the mental patient seeks medical treatment difficult to surmount. Some patients and their families refuse to recognize depression as an illness and seek proper treatment for fear of being stigmatized as "mentally ill" and of not being able to hold their heads up in the future. This brings great trouble to the treatment of mental illness and often delays the treatment.

  1. Methods section

    Structural equation modeling is recommended, which is better than the method currently used by the authors.

  1. Discussion section

Further discussion and interpretation of the relationship between competency to consent to treatment and psychological well-being in Psychiatric patient is necessary.

  1. Reference

The reference section is too old, please cite more references in the last 3 years.

Author Response

Dear. Reviewer 1.

Thank you for your review.
Unfortunately, in the case of the details of the review evaluation, it was uploaded as an attachment due to the cunning and a lot of content. Please confirm.

I would like to express my gratitude to the attentive and meticulous couple.

Best Regard

from. Dr. hur

Reviewer 3 Report

It is necessary to know the total number of the population, which could have participated. It is recommended to use more up-to-date bibliographic references.

Author Response

Dear. Reviewer 2.

Thank you for your review.
Unfortunately, in the case of the details of the review evaluation, it was uploaded as an attachment due to the cunning and a lot of content. Please confirm.

I would like to express my gratitude to the attentive and meticulous couple.

Best regards
from. Dr. Hur

Round 2

Reviewer 1 Report

Generally speaking, I think most comments have been responded, and the use of figures is more efficient when interpreting the results. However, I insist that the document must be re-read by the authors to eliminate repeated contents, and to make sentences and paragraphs more attractive for readers.

Please, check this sentence from the abstract: “The study participants consisted of 191 people with mental disorders voluntarily consenting to the study among mentally disabled people receiving outpatient treatment of the psychiatric outpatient was found to be good overall as the average of both the overall and the sub-factors, ability of”. It seems like it is lacking punctuation, and I cannot understand what you are trying to say.

The following sentence does not serve any purpose in the abstract: “The researchers of this study checked the contents and deleted the duplicate contents.”

2.1: “After excluding data of five patients who responded insincerely, (e.g. unified answer with one number)”. This still makes no sense to me. What do you mean “one number”? This was better explained in the cover letter of your responses.

Regarding the Procedure and Analysis method, shouldn’t they be numbered in their subheadings?

Lines 330-332 repeat the main objective, please be consistent, there is no need to establish it everywhere.

Author Response

Dear. Reviewer 1.

We thank you for your meticulous and thorough review. We conducted a review with the researchers about the content you sent.

Edited and the modified file is attached separately In addition, the answer was also written and attached as a file in the same way as before.

Thank you again.
Best Regard

from. Dr. Hur
